# Four Novel SNPs of *MYO1A* Gene Associated with Heat-Tolerance in Chinese Cattle

**DOI:** 10.3390/ani9110964

**Published:** 2019-11-13

**Authors:** Peng Jia, Cuicui Cai, Kaixing Qu, Ningbo Chen, Yutang Jia, Quratulain Hanif, Jianyong Liu, Jicai Zhang, Hong Chen, Bizhi Huang, Chuzhao Lei

**Affiliations:** 1Key Laboratory of Animal Genetics, Breeding and Reproduction of Shaanxi Province, College of Animal Science and Technology, Northwest A&F University, Yangling 712100, China; nwafujp@gmail.com (P.J.); caicuicui3c@163.com (C.C.); ningboch@126.com (N.C.); chenhong1212@263.net (H.C.); 2Yunnan Academy of Grassland and Animal Science, Kunming 650212, China; kaixqu@163.com (K.Q.); ljy@ynbp.cn (J.L.); ynzjc@126.com (J.Z.); 3Institute of Animal Science and Veterinary Medicine, Anhui Academy of Agriculture Science, Hefei 230001, China; yutang2018@163.com; 4National Institute for Biotechnology and Genetic Engineering, Pakistan Institute of Engineering and Applied Sciences, Faisalabad 577, Pakistan; micro32uvas@gmail.com

**Keywords:** Chinese cattle, *MYO1A* gene, SNP, heat-tolerance trait, association

## Abstract

**Simple Summary:**

Chinese indigenous cattle were clustered into two groups of southern and northern breeds. The populations distributed in the hot south are mainly dominated by *Bos indicus*, and the populations distributed in the cold north are mainly dominated by *Bos taurus*. Myosin-1a (*MYO1A*) gene is a candidate gene related to pigmentation, which may be closely related to the heat tolerance traits of cattle. Therefore, our study aimed to explore the relationship between *MYO1A* gene and heat tolerance in Chinese indigenous cattle.

**Abstract:**

Based on the previous GWAS research related to bovine heat tolerance trait, this study aimed to explore the effect of myosin-1a (*MYO1A*) gene on bovine heat tolerance trait, and find the molecular markers related to the heat tolerance of Chinese cattle. In our study, four novel candidate SNPs highly conserved in *B. indicus* breeds but barely existed in *B. taurus* were identified in *MYO1A* gene according to Bovine Genome Variation Database and Selective Signatures (BGVD). PCR and DNA sequencing were used to genotype 1072 individuals including 34 Chinese indigenous cattle breeds as well as Angus and Indian zebu. Two synonymous mutations (rs208210464 and rs110123931), one missense mutation (rs209999142; Phe172Ser), and one intron mutation (rs135771836) were detected. The frequencies of mutant alleles of the four SNPs gradually increased from northern groups to southern groups of Chinese cattle, which was consistent with the distribution of various climatic conditions of China. Additionally, four SNPs were significantly associated with four climatic conditions including annual mean temperature (T), relative humidity (H), temperature-humidity index (THI), and average annual sunshine hours (100-cloudiness) (SR). Among these, rs209999142 and Hap 1/1 had better performance than others. Our results suggested that rs209999142 was associated with heat-tolerance trait and rs208210464, rs110123931, and rs135771836 showed high phenotypic effect on heat-tolerance trait because of the strong linkage with rs209999142. These SNPs could be used as candidates for marker-assisted selection (MAS) in cattle breeding.

## 1. Introduction

Metabolic disorders, loss of appetite, slow growth, and impaired immune function may occur under heat stress, which would cause serious economic issue [1]. As the frequency and duration of heat-stress conditions increasing, projected future increases in temperature could have significant negative implications on the productivity and welfare of ruminants that play a significant role in terms of world food production [2]. Although advances in environmental cooling systems ameliorate production losses during summer months, heat stress continues to cost a lot annually. Increase in solar radiation can cause the surrounding air to warm up, which could create a heat-stress problem. Heat exchange by radiation depended upon the reflective properties of the hair coat. Light-colored hair coats reflected a greater proportion of incident solar radiation than hair coats that were dark in color [3]. Hence, cattle exposed chronically to high temperature became progressively lighter in color and there was some evidence that this change occurred more rapidly for Brahman than Brown Swiss or Jersey [4].

Family members of the myosin superfamily are widely expressed and tightly integrated in cellular networks of interlinked biochemical pathways where they function as integrators between signaling and the dynamics of cytoskeletal mechanics [5,6]. Class-1 myosins are single-headed and dynamically link the actin cytoskeleton to membranes, spend only a small fraction of time attached to actin [7]. This feature does not allow them to transport cargoes but instead to function as strain sensors at the cytoskeleton-plasma membrane interface, a function essential to the regulation of membrane tension and cell migration [7]. It suggests myosin-1 isoforms may be involved in symmetry breaking events. Processivity which is facilitated by a high duty ratio is a prerequisite to transport cargoes such as melanosomes or endosomes along the actin cytoskeleton over long distances to its cellular destination [8]. *MYO1A* gene is one member of Class-1 myosins. In line with the previous genomic SNP array data, *MYO1A* was considered to be a candidate gene related to pigmentation in cattle [9]. Then we identified a fragment with four novel SNPs in *MYO1A* gene in cattle by BGVD (http://animal.nwsuaf.edu.cn/code/index.php/BosVar) [10], which were highly conserved in *B. indicus* breeds but barely existed in *B. taurus*, may be candidate SNPs associated with adaptation under heat-stress condition. BGVD is a platform based on a large number of resequencing data of cattle, through which we can observe the expected distribution of SNPs in groups of dozens of different cattle breeds.

China exhibits five climatic zones, with 53 Chinese cattle breeds and rich genetic resources in this specific environment, and the thermal conditions are gradually increasing from north to south [11]. Chinese indigenous cattle originated from *Bos taurus* and *Bos indicus*, were clustered into two groups of southern and northern breeds [12]. Under a long-term introgression and selection between southern breeds and northern breeds, a distinct transition occurred from northern to southern, which has been demonstrated by genetic studies on Y chromosome polymorphisms, mitochondrial DNA markers, and whole-genome scan data [12,13,14,15].

Consequently, Chinese indigenous cattle breeds are suitable resources to detect SNPs in the bovine *MYO1A* gene and for testing the gene possible association with the heat tolerance trait. It is expected that the results of this study can be used for heat tolerance breeding.

## 2. Materials and Methods

All experiments performed in this study were approved by the International Animal Care and Use Committee of the Northwest A&F University (IACUC-NWAFU). Furthermore, the care and use of animals were fully compliant with local animal welfare laws, guidelines, and policies.

### 2.1. DNA Samples and Data Collection

Genomic DNA of 1072 individuals from 36 breeds including 34 Chinese indigenous cattle breeds as well as Angus (*B. taurus*) and Indian zebu (*B. indicus*) (Appendix A) were isolated from ear tissues by the standard phenol-chloroform method [16]. DNA samples were diluted to a standard concentration (50 ng µL^−1^) and stored at −80 °C.

Geographical data over the last 30 years for the sampling sites of the 34 indigenous cattle breeds were collected from the Chinese Central Meteorological Office (http://data.cma.cn) (Appendix A) and were used to estimate heat tolerance traits.

### 2.2. SNP Detecting and Genotyping

The polymerase chain reaction (PCR) primer was designed based on the bovine *MYO1A* sequence (GenBank accession no. NC_037332.1) by the Primer Premier 5.0 (Premier Biosoft International, Palo Alto, CA, USA). The forward primer was 5′-AAGACCATTCGCAATAACA-3′ and the reverse primer was 5′-TTCCTCCTTCCTACAGCA-3′. PCR product size was 318 bp. PCR amplification system was performed in a 20 μL volume system: 50 ng genomic, 10 μL 2× PCR mix, 0.5 μM of each primers and 8 μL ddH_2_O. The cycling protocol was as follows: denaturation for 5 min at 95 °C; 35 cycles of 94 °C for 30 s, annealing at 51 °C for 30 s, primer extension at 72 °C for 30 s; with a final extension performed at 72 °C for 8 min. The PCR products were detected by electrophoresis on a 1% agarose gel stained with ethidium bromide and sequenced directly at Shanghai Sangon Biotech Company, Shanghai, China. Sequencing results were analyzed with SEQMAN TMIIv 6.1 (DNASTAR, Inc., Madison, WI, USA).

### 2.3. Statistical Analysis

SHEsis software was used to perform haplotype analysis [17]. Codon bias of synonymous mutations was analyzed by online software Countcodon program (version 4) (http://www.kazusa.or.jp/codon/countcodon.html). Homology modeling of proteins before and after missense mutation was performed based on the bovine *MYO1A* sequence (NP_776820.1) by SWISS-MODEL online software (https://swissmodel.expasy.org/) [18].

McDowell et al. [19] suggested that temperature-humidity index (THI) can be used as an indicator to ascertain heat load intensity of thermal climatic conditions by measuring the combined effects of annual mean temperature (T) and relative humidity (H). Solar radiation was affected by solar altitude angle, local climate types, and solar radiation time, which had the largest influence (r^2^ = 0.92), and solar radiation time basically reflected the solar radiation amount [20]. In consequence, T, H, THI, and average annual sunshine hours (100-cloudiness) (SR) were used for association analysis with genotypes or haplotypes by SPSS 18.0 software (SPSS, Inc., Armonk, NY, US). THI was calculated based on the formula used by the National Oceanic and Atmospheric Administration [21]: THI = (1.8T + 32) − (0.55 − 0.0055H) (1.8T − 26), where T was temperature in degrees Celsius and H was relative humidity as a percentage. The statistical linear model was used: Y_i_ = µ + G_i_ + B_i_ + e_i_, where Y_i_ was the value of T, H, THI, and SR between 1951 and 1980; µ was the mean value; G_i_ was the fixed effect of the genotypes or haplotypes; B_i_ was the fixed effect of breeds; e_i_ was the random residual effect.

## 3. Results

### 3.1. SNPs Screening and Genotyping

In our study, four novel SNPs were detected in bovine *MYO1A* gene: rs208210464 (g.56383560G > A), rs209999142 (g.56383565T > C), rs110123931 (g.56383578T > C), and rs135771836 (g.56383635A > G). Among the four mutations, rs208210464 and rs110123931 were synonymous mutations. Besides, rs209999142 was a missense mutation which causes phenylalanine to serine amino acid substitution, and rs135771836 located in the intron region (Figure 1). A distinct transition can be observed from northern breeds to southern breeds (Figure 2, Appendix A). Furthermore, frequencies of all mutant genotypes in Zebu population were more than 0.68, whereas frequencies of all wild genotypes in Angus population were more than 0.84 (Appendix A).

### 3.2. Haplotype Analysis and Linkage Disequilibrium

To reveal the linkage relationships among rs208210464, rs209999142, rs110123931, and rs135771836, the linkage disequilibrium (LD) between the four mutations were estimated (Figure 3). D′ values ranged from 0.90 to 0.99. r^2^ values ranged from 0.28 to 0.96 (Table 1). Due to r^2^ > 0.33 was the indicative of strong linkage disequilibrium [22], strong linkage between all SNPs were revealed except rs208210464 and rs110123931 (r^2^ = 0.29) and rs110123931 and rs135771836 (r^2^ = 0.28). Besides, rs208210464 almost completely linked to rs135771836 (r^2^ = 0.96). The haplotype analysis showed that four different haplotypes were identified among the four SNPs (all those frequencies < 0.05 had been ignored in analysis). Hap 1 had the highest haplotype frequency (0.39), followed by Hap 4 (0.29), Hap 2 (0.15), and Hap 3 (0.14) (Table 2).

### 3.3. Association Analysis of SNPs and Haplotype Combination

The results of the association analysis between the four SNPs and the four environmental parameters (T, H, THI, and SR) for 34 breeds in 1006 Chinese indigenous cattle were shown in Table 3. For rs135771836, there was no significant difference between individuals with genotype GG and AG with SR. For rs208210464, individuals with genotype GG and AA were significantly different with SR (*p* < 0.05). For rs110123931, individuals with genotype CT and TT were significantly different with H (*p* < 0.05). In addition to the above, there were significant differences among different genotypes for all SNPs with all environmental parameters (T, H, THI, and SR) (*p* < 0.01).

The results of the association analysis between the ten combined haplotypes and the four environmental parameters (T, H, THI, and SR) for 34 breeds in 956 Chinese indigenous cattle were shown in Table 4. According to Table 4, Hap 1/1 was the most potentially related to heat tolerance traits. Tests of effects of the four environmental parameters on the *MYO1A* genotypes indicated that annual average temperature had the strongest correlation with the genotypes (Appendix A).

### 3.4. Mutation Analysis

Among the four novel SNPs, rs208210464 and rs110123931 were synonymous mutations, rs209999142 was a missense mutation. rs209999142 was predicted the protein three-dimensional structure by the homologous modelling method with SWISS-MODEL software (Figure 4). There was a significant difference in the value of solvation between wild type and mutant type (Table 5), which meant wild type might had a stronger residue embedding ability than mutant type. Additionally, the conversion of phenylalanine at the 172 site to serine led to the rise of the value of qualitative model energy analysis (QMEAN) from 0.73 to 0.79. Genetic codons before and after mutation was shown in Figure 5, and there were some differences before and after two synonymous mutations for codon bias, which may cause functional differences.

## 4. Discussion

With the advent of escalating environmental stress due to climate change, the challenging tropical stress could have determine more favourable genotypes for bovine [23]. Nevertheless, as the environments and management intervention strategies implementation, environmental regulation removed environmental stressors, this would make natural selection be adversely affected. In consequence, screening of candidate genes related to heat tolerance and breeding of heat-resistant cattle is of great significance to the production of cattle.

*MYO1A*, also known as “brush border myosin-I”, has been identified as a cause of deafness and had tumor suppressor activity in the intestine [24,25]. Nonetheless, *MYO7A* and *MYO5A*, homologous genes of *MYO1A*, have been previously found to play an important role in melanin transport [26,27]. MYO5A is a major actin-based vesicle transport motor that binds to one of its cargos, the melanosome, by means of a RAB27A/MLPH receptor [27]. It is essential for the capture and local transport of melanosomes in dendritic tips, leading to a dilution of coat color after mutated [28]. MYO7A, which transports melanosome along the apical processes, may play a similar transport role in the pigmented retina [29]. The genes from same family have obvious similarity in structure and function, and encodes similar protein products. Through 3D modeling of *MYO1A* gene, we found that it has high sequence identity with *MYO7A* (39.14%) and *MYO5A* (40.27%), respectively. Therefore, we suggested that *MYO1A* has similar functions as *MYO7A* and *MYO5A* gene, melanosomes transport. Moreover, earlier reports found that *MYO1A* gene in the region of bovine chromosome 5 was suggested to have an effect on pigmentation in an F_2_-Backcross Charolais × Holstein population by a linkage mapping study [30]. Edea et al. [9] revealed that *MYO1A* gene related to coat color may have contributed to adaptation under tropical conditions by 80K Indicine BeadChip scan. Thus, it is the first study on the relationship between *MYO1A* gene and heat-tolerance.

The vast varieties in longitude, latitude, and altitude form the extremely complex climate of China. The northern part of the coastal zone has a temperate monsoon climate, whereas the south has a subtropical monsoon climate. The Qinghai-Tibet Plateau is a plateau climate, and other inland areas are basically temperate continental climates. In Figure 2, the mutant allele’s frequencies of the four SNPs we detected had an extremely regular increasing trend from the south to the north. This trend was consistent with the distribution of different climatic conditions in China (Appendix A). It also shows an obvious distribution trend from south to north. This confirmed with the fact that southern and northern breeds were affected by the introgression of heat-resistant *Bos indicus* and heat-sensitive *Bos taurus*, respectively [31].

According to Table 3, rs209999142, the missense mutation, showed stronger association with four climatic conditions than other mutations, which may be due to weakening of residue embedding ability and improvement of QMEAN. The synonymous mutations (rs208210464 and rs110123931) led the changes of codon bias, these changes may affect the efficiency of protein synthesis in the process of translation and thus the normal expression of function. As for rs135771836 (intron mutation), the association between this mutation and thermal adaptation may be due to the high strength linkage with mutation rs208210464 in exon. Combining the results of LD and association analysis, we suggested that rs209999142 was a causative variant and the others showed association with heat-tolerance trait due to the strong linkage with rs209999142. Hap1 had the highest haplotype frequency (0.39), and high-frequency haplotypes had been probably present in the populations for a long time, which may be directly or indirectly influenced by different rearing environments [32]. Combined with the results of combined haplotypes association analysis (Table 4), Hap 1/1 was the most possibly associated with heat-tolerance trait.

## 5. Conclusions

In summary, the identification of four different loci suggested that this gene may be a significant contributor to bovine heat-tolerance trait. Additional studies were required to definitively prove the real contribution of *MYO1A* to heat-tolerance in tissue and cell level. Furthermore, rs209999142 and Hap 1/1 could be used in marker-assisted selection (MAS) programs to improve heat tolerance trait in Chinese cattle.

## Figures and Tables

**Figure 1 animals-09-00964-f001:**
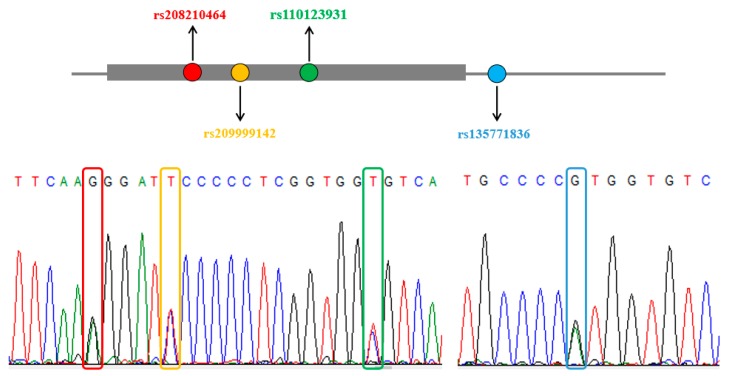
Loci and sequencing maps of the four SNPs in bovine *MYO1A* gene.

**Figure 2 animals-09-00964-f002:**
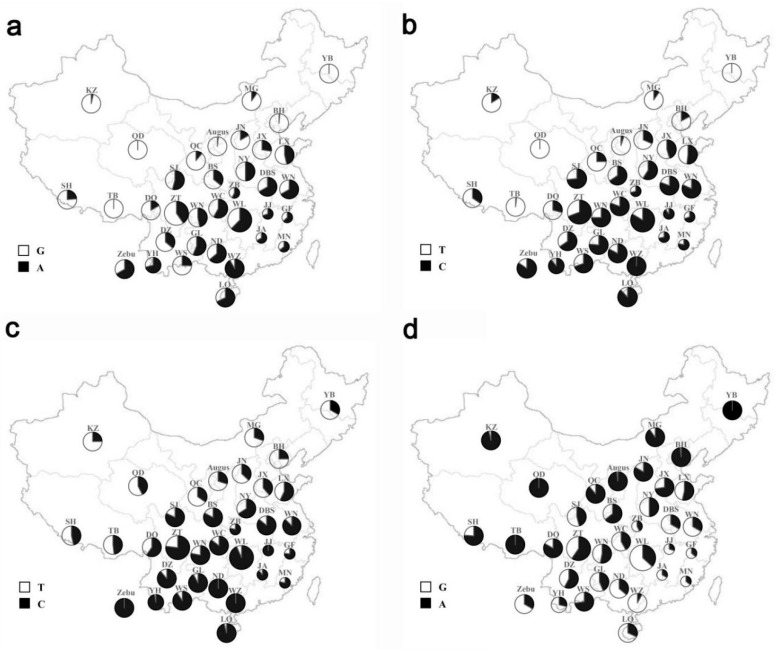
Geographical distribution of four variants among 34 Chinese breeds as well as Augus and Indian zebu populations. (**a**–**d**) showed the distributions of different alleles of rs208210464, rs209999142, rs110123931, and rs135771836, respectively. BH, Bohai Black; BS, Bashan; DBS, Dabieshan; DZ, Dianzhong; DQ, Diqing; GF, Guangfeng; GL, Guanling; GZWN, Weining; JA, Ji’an; JJ, Jinjiang; JN, Jinnan; JX, Jiaxian red; KZ, Kazakh; LQ, Leiqiong; LX, Luxi; LL, Longlin; MG, Mongolian; MN, Minnan; NY, Nanyang; ND, Nandan; QC, Qinchuan; QD, Qaidam; SJ, Sanjiang; SH, Shigatse Humped; TB, Tibetan; WC, Wuchuan; WL, Wuling; WN, Wannan; WS, Wenshan; WZ, Weizhou; YB, Yanbian; YH, Yunnan Humped; ZB, Zaobei; ZT, Zhaotong.

**Figure 3 animals-09-00964-f003:**
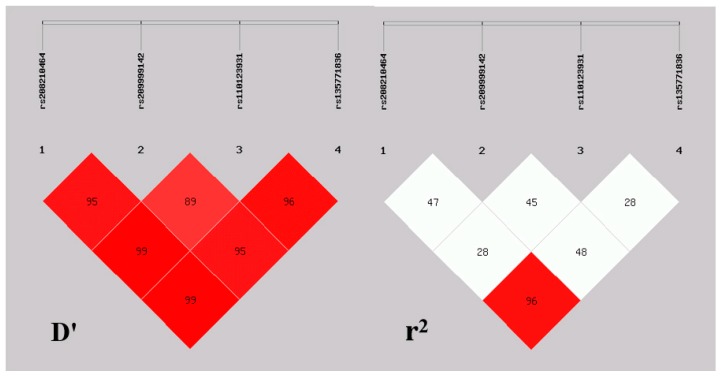
Linkage disequilibrium plot of four loci in bovine *MYO1A* gene.

**Figure 4 animals-09-00964-f004:**
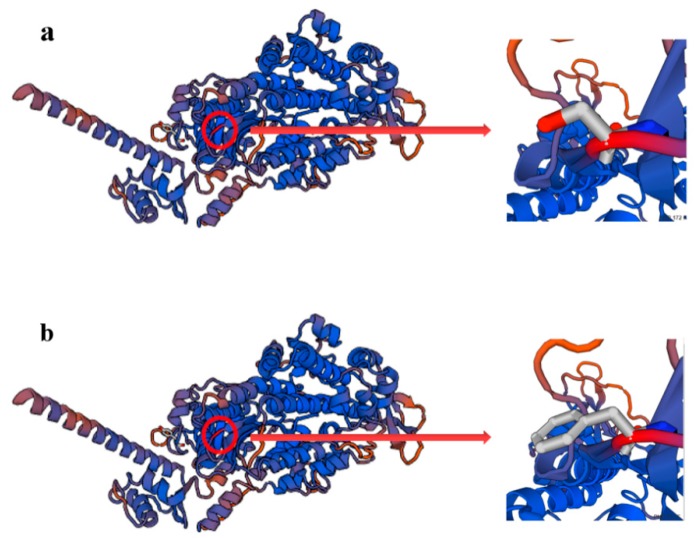
Three-dimensional structure prediction of bovine MYO1A protein. (**a**) Mutant type (phenylalanine); (**b**) Wild type (serine).

**Figure 5 animals-09-00964-f005:**
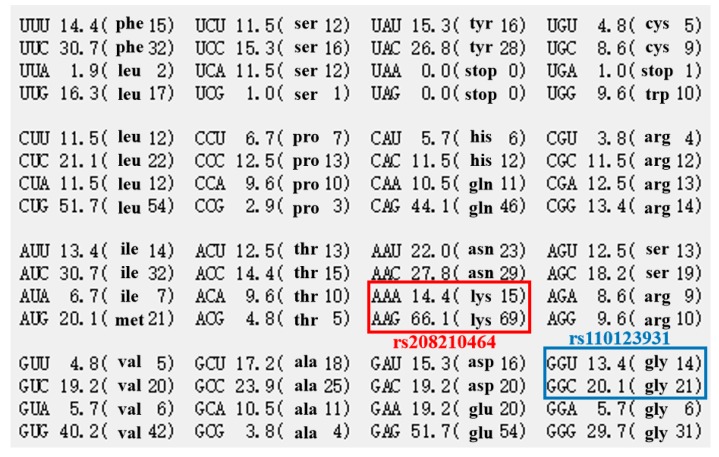
Distribution of genetic codons in bovine *MYO1A* gene. Note: the red and blue frames represent amino acid preference of codons before and after mutation (rs208210464 and rs110123931), respectively.

**Table 1 animals-09-00964-t001:** Estimated values of linkage disequilibria for four SNPs in *MYO1A* gene.

D′/r^2^	rs209999142	rs110123931	rs135771836
rs208210464	0.96/0.48	0.99/0.29	0.99/0.96
rs209999142	-	0.90/0.46	0.95/0.49
rs110123931	-	-	0.97/0.28

**Table 2 animals-09-00964-t002:** Haplotypes of the four SNPs in bovine *MYO1A* gene.

Haplotype	rs208210464	rs209999142	rs110123931	rs135771836	Frequency
Hap1	A	C	C	G	0.39
Hap2	G	T	T	A	0.29
Hap3	G	C	C	A	0.15
Hap4	G	T	C	A	0.14

All those frequencies < 0.05 had been ignored in analysis.

**Table 3 animals-09-00964-t003:** Associations of four SNPs genotypes of *MYO1A* gene with annual mean temperature (T), relative humidity (H), temperature-humidity index (THI), average annual sunshine hours (SR).

Loci	Genotype (n)	T (°C) (LSM ± SE)	H (%) (LSM ± SE)	THI (LSM ± SE)	SR (%) (LSM ± SE)
rs208210464 (g.56383560 G > A)	AA (222)	17.05 ± 0.30 ^A^	76.33 ± 0.49 ^A^	61.06 ± 0.41 ^A^	42.79 ± 0.60 ^c^
	AG (359)	13.93 ± 0.26 ^B^	72.43 ± 0.46 ^B^	57.20 ± 0.38 ^B^	44.90 ± 0.54 ^B^
	GG (425)	10.39 ± 0.25 ^C^	65.08 ± 0.58 ^C^	52.42 ± 0.35 ^C^	53.14 ± 0.53 ^a^
rs209999142 (g.56383565 T > C)	CC (413)	16.19 ± 0.23 ^A^	75.69 ± 0.38 ^A^	60.06 ± 0.33 ^A^	42.89 ± 0.46 ^C^
	CT (306)	13.20 ± 0.27 ^B^	70.80 ± 0.48 ^B^	56.26 ± 0.40 ^B^	47.00 ± 0.59 ^B^
	TT (285)	8.59 ± 0.26 ^C^	61.55 ± 0.71 ^C^	49.95 ± 0.36 ^C^	56.22 ± 0.61 ^A^
rs110123931 (g.56383578 T > C)	CC (553)	15.62 ± 0.21 ^A^	74.87 ± 0.36 ^A^	59.36 ± 0.30 ^A^	43.81 ± 0.42 ^C^
	CT (288)	10.67 ± 0.28 ^B^	65.53 ± 0.68 ^b^	52.85 ± 0.40 ^B^	50.89 ± 0.63 ^B^
	TT (165)	9.02 ± 0.32 ^C^	62.60 ± 0.73 ^c^	50.46 ± 0.24 ^C^	56.46 ± 0.77 ^A^
rs135771836 (g.56383635 A > G)	AA (417)	10.22 ± 0.25 ^C^	64.81 ± 0.52 ^C^	52.17 ± 0.34 ^C^	53.18 ± 0.54 ^A^
	AG (362)	13.95 ± 0.26 ^B^	72.47 ± 0.45 ^B^	57.24 ± 0.37 ^B^	44.90 ± 0.54 ^b^
	GG (227)	17.13 ± 0.30 ^A^	76.41 ± 0.48 ^A^	61.25 ± 0.41 ^A^	43.04 ± 0.60 ^b^

^a, b, c^ indicate significant differences between genotypes with *p* < 0.05. ^A, B, C^ indicate significant differences between genotypes with *p* < 0.01.

**Table 4 animals-09-00964-t004:** Associations of combined haplotypes of *MYO1A* gene with annual mean temperature (T), relative humidity (H), temperature-humidity index (THI), average annual sunshine hours (SR).

Combined Haplotypes	T (°C) (LSM ± SE)	H (%) (LSM ± SE)	THI (LSM ± SE)	SR (%) (LSM ± SE)
Hap 1/1 (0.22)	16.98 ± 0.31 ^a^	76.29 ± 0.50 ^a^	60.9 ± 0.42 ^a^	42.83 ± 0.61 ^D^
Hap 1/3 (0.16)	15.23 ± 0.40 ^b^	74.96 ± 0.65 ^a^	58.8 ± 0.56 ^a^	42.67 ± 0.80 ^D^
Hap 2/2 (0.15)	8.38 ± 0.31 ^D^	61.69 ± 0.78 ^C^	49.5 ± 0.43 ^C^	57.60 ± 0.79 ^a^
Hap 1/2 (0.13)	12.00 ± 0.42 ^C^	68.88 ± 0.74 ^B^	54.5 ± 0.61 ^B^	48.12 ± 0.89 ^c^
Hap 2/4 (0.10)	7.88 ± 0.41 ^D^	59.33 ± 1.44 ^C^	49.1 ± 0.56 ^C^	55.81 ± 1.10 ^a^
Hap 1/4 (0.07)	14.66 ± 0.58 ^b^	73.87 ± 1.05 ^a^	58.3 ± 0.84 ^a^	42.88 ± 1.32 ^D^
Hap 2/4 (0.05)	12.25 ± 0.65 ^C^	68.72 ± 1.03 ^B^	54.9 ± 0.95 ^B^	48.56 ± 1.37 ^c^
Hap 4/4 (0.04)	10.75 ± 1.03 ^C^	66.28 ± 2.24 ^B^	53.0 ± 1.46 ^B^	53.31 ± 1.80 ^b^
Hap 3/3 (0.04)	15.37 ± 0.93 ^a^	75.14 ± 1.47 ^a^	59.1 ± 1.35 ^a^	42.79 ± 1.72 ^D^
Hap 3/4 (0.03)	14.82 ± 0.79 ^b^	74.81 ± 1.22 ^a^	58.3 ± 1.14 ^a^	46.33 ± 1.73 ^c^

^a, b, c^ indicate significant differences between haplotypes with *p* < 0.05. ^A, B, C, D^ indicate significant differences between haplotypes with *p* < 0.01.

**Table 5 animals-09-00964-t005:** Analysis and contrast table for each parameter of the three-dimensional structure.

Type	Seq Identity	QMEAN	C-Beta	All Atom	Solvation	Torsion
Wild type	66.53%	2.98	1.71	0.56	0.66	2.89
Mutant type	66.53%	2.99	1.71	0.55	0.63	2.89

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
