# Peer review of "Four Novel SNPs of MYO1A Gene Associated with Heat-Tolerance in Chinese Cattle"

_animals, 2019, doi:10.3390/ani9110964_

Round 1

Reviewer 1 Report

The manuscript has been corrected according to all comments. I recommend this study for publication in Animals journal. 

Author Response

Dear reviewer

Thank you very much for your suggestions on my manuscript.

Reviewer 2 Report

Heat stress seriously affects the production performance of cattle.In the breeding process of cattle, how to improve the heat tolerance ability of cattle is an important aspect of the breeding of high-quality beef cattle.The research of Chuzhao Lei et al. is aimed at exploring SNPS in MYO1A gene related to heat tolerance. It provides important molecular markers for cattle breeding in the future. But this article has a few corrections as follows: 1. In the section of materials and methods, the temperature, humidity and other data shall be specified in detail. How does this part of temperature and humidity data correspond to individuals? Usually, studies on heat tolerance are conducted to test the rectal temperature and blood indexes of individuals under the same condition of heat stress to reflect the resistance of cattle to heat tolerance, and then the genetic background is combined with these date for analysis. 2. How does SNP in MYO1A gene affect the ability of cattle to resist heat stress? Is there a difference of this gene expressed in different cattle (breeds) that are resistant to heat stress? If so, the study is more convincing.

Author Response

Dear reviewer

Thank you very much for your suggestions on my manuscript. For your question, my reply is as follows:

In the section of materials and methods, the temperature, humidity and other data shall be specified in detail. How does this part of temperature and humidity data correspond to individuals? Usually, studies on heat tolerance are conducted to test the rectal temperature and blood indexes of individuals under the same condition of heat stress to reflect the resistance of cattle to heat tolerance, and then the genetic background is combined with these date for analysis.

Heat stress refers to the imbalance of the ability of animals to regulate under harsh conditions. This study is devoted to exploring the adaptive evolution of cattle under the pressure of thermal environment. Therefore, environmental data is selected as an indicator for evaluating thermal adaptability. Moreover, our research has a large number of samples and a wide sampling range, so the results are trustworthy. The same research method can be seen in the following articles:

Zeng, L. , Chen, N. , Ning, Q. , Yao, Y. , Chen, H. , Dang, R. , Zhang, H. and Lei, C. (2018), PRLH and SOD1 gene variations associated with heat tolerance in Chinese cattle. Anim Genet, 49: 447-451. doi:10.1111/age.12702

Wang, K., Cao, Y., Rong, Y., Ning, Q., Jia, P., Huang, Y., … Lei, C. (2019). A Novel SNP in EIF2AK4 Gene Is Associated with Thermal Tolerance Traits in Chinese Cattle. Animals : an open access journal from MDPI, 9(6), 375. doi:10.3390/ani9060375

Eckert A.J., Joost V.H., Wegrzyn J.L., C Dana N., Jeffrey R.I., González-Martínez S.C. & Neale D.B. (2010) Patterns of population structure and environmental associations to aridity across the range of loblolly pine (Pinus taeda L., Pinaceae). Genetics 185, 969.

Hancock A.M., Benjamin B., Nathalie F., Horton M.W., Jarymowycz L.B., F Gianluca S., Chris T., Fabrice R. & Joy B. (2011) Adaptation to climate across the Arabidopsis thaliana genome. Science 334, 83-6.

How does SNP in MYO1A gene affect the ability of cattle to resist heat stress? Is there a difference of this gene expressed in different cattle (breeds) that are resistant to heat stress? If so, the study is more convincing.

Thanks, it is a good idea. However, different samples of cattle tissues were not collected before and Chinese cattle distribute in vast area. It is a huge work to collect so many samples within few days, so we cannot add this experiment in this study.

Round 2

Reviewer 2 Report

Our question has been answered.

Author Response

Dear reviewer

Thank you very much for your suggestions on my manuscript.

This manuscript is a resubmission of an earlier submission. The following is a list of the peer review reports and author responses from that submission.

Round 1

Reviewer 1 Report

The manuscript contains English errors, which should be corrected through the manuscript. You used colloquial wording and the same sentences are hard to follow, examples which need corrections: 182 ‘obvious differenses’ – what does it mean? It is not a scientific term. 210 – has – have 213 – ‘belived’? here, it is a colloquial world. please change on ‘suggested’ 215 – was – were

In my opinion there is a lack of more information about why exactly these polymorphisms were analyzed. Does analyzed part of MYO1A gene is associated with gene expression or important protein part? How do SNPs can affect protein function? Is should be explained wider.

Abstract lines 34-37 -  I think that these conclusions are too far-reaching. Due to the fact that your results confirmed high LD between SNPs you can not present that all 4 SNPs are associated with  hest-tolerance traits. I suggest to clearly show that one of presented mutation is a causative variant and the others ar linkage with such SNP with high phenotypic effect and thus association was obtained for all polymorphisms.  This important issue should also be further discussed in Discussion section.

Line 127-129:  This statement is not clear, plese describe more clearly – it is hard to follow which rs you described.

L 157-163: Did you check breed effect on investigated traits?

185-187: It should be explained why showed synonymous SNPs can cause functional differences.

Table 2.  I suggested to change the order od presented haplotypes according to frequency.

Table 4. Please add the frequency of obtained diplotypes. It will be easier to reading and analyzing data.

Figure 4 – The footnote of this figure should be improved and more detail should be described about possible effect of given mutations. The accession number of protein sequence should be presented.

Author Response

Dear reviewer

Thank you very much for your suggestions on my manuscript. For your question, my reply is as follows:

Point 1: The manuscript contains English errors, which should be corrected through the manuscript. You used colloquial wording and the same sentences are hard to follow, examples which need corrections:

182 ‘obvious differenses’ – what does it mean? It is not a scientific term.

210 – has – have     213 – ‘belived’? here, it is a colloquial world. please change on ‘suggested’         215 – was – were

Response 1:

Line 182: There was a significant difference in the value of solvation between wild type and mutant type...

Line 210: The genes from same family have obvious similarity in structure and function, and encodes similar protein products.

Line 213: Therefore, we suggested that MYO1A has similar functions as MYO7A and MYO5A gene, melanosomes transport.

Line 215: Moreover, earlier reports found that MYO1A gene in the region of bovine chromosome 5 flanked by markers ETH10 and DIK5248 was suggested to have an effect on pigmentation.

Point 2: In my opinion there is a lack of more information about why exactly these polymorphisms were analyzed. Does analyzed part of MYO1A gene is associated with gene expression or important protein part? How do SNPs can affect protein function? Is should be explained wider.

Response 2:

In our study, four novel candidate SNPs highly conserved in B. indicus breeds but barely existed in B. taurus were identified in MYO1A gene according to Bovine Genome Variation Database and Selective Signatures (BGVD). BGVD is an early achievement of our laboratory. It is a platform based on a large number of resequencing data of cattle, through which we can observe the expected distribution of SNPs in groups of dozens of different cattle breeds. And we have added in the introduction: “BGVD is a platform based on a large number of resequencing data of cattle, through which we can observe the expected distribution of SNPs in groups of dozens of different cattle breeds”

The analysis did not involve the verification of the actual expression function of the MYO1A protein, but only the prediction of the protein model and codon bias of synonymous mutations. The missense mutation didn't locate in important protein part. And we have added in the discussion: ”And the synonymous mutations (rs208210464 and rs110123931) led the changes of codon bias, these changes may affect the efficiency of protein synthesis in the process of translation and thus the normal expression of function. As for rs135771836 (intron mutation), the association between this mutation and thermal adaptation may be due to the high strength linkage with mutation rs208210464 in exon. ”

Point 3: Abstract lines 34-37 -  I think that these conclusions are too far-reaching. Due to the fact that your results confirmed high LD between SNPs you can not present that all 4 SNPs are associated with  hest-tolerance traits. I suggest to clearly show that one of presented mutation is a causative variant and the others are linkage with such SNP with high phenotypic effect and thus association was obtained for all polymorphisms.  This important issue should also be further discussed in Discussion section.

Response 3:  

We have modified in the abstract: “Our results suggested that rs209999142 was associated with heat-tolerance trait and rs208210464, rs110123931 and rs135771836 showed high phenotypic effect on heat-tolerance trait because of the strong linkage with rs209999142.” And we have added in the discussion: ”Combining the results of LD and association analysis, we suggested that rs209999142 was a causative variant and the others showed association with heat-tolerance trait due to the strong linkage with rs209999142. ”

Point 4: Line 127-129:  This statement is not clear, plese describe more clearly – it is hard to follow which rs you described.

Response 4:  

Modified as: “Among the four mutations, rs208210464 and rs110123931 were synonymous mutations. Besides, rs209999142 was a missense mutation which causes phenylalanine to serine amino acid substitution, and rs135771836 located in the intron region.”

Point 5: L 157-163: Did you check breed effect on investigated traits?

Response 5:  

The fixed effect of breeds has been counted in the statistical linear model (Line117-120).

Point 6: 185-187: It should be explained why showed synonymous SNPs can cause functional differences.

Response 6:  

We have added in the discussion: ”And the synonymous mutations (rs208210464 and rs110123931) led the changes of codon bias, these changes may affect the efficiency of protein synthesis in the process of translation and thus the normal expression of function. As for rs135771836 (intron mutation), the association between this mutation and thermal adaptation may be due to the high strength linkage with mutation rs208210464 in exon. ”

Point 7: Table 2.  I suggested to change the order od presented haplotypes according to frequency.

Response 7:  

The order has been changed.(Table 2. and Table 4.)

Point 8: Table 4. Please add the frequency of obtained diplotypes. It will be easier to reading and analyzing data.

Response 8:  

The frequency of obtained diplotypes has been added.(Table 4.)

Point 9: Figure 4 – The footnote of this figure should be improved and more detail should be described about possible effect of given mutations. The accession number of protein sequence should be presented.

Response 9:  

The parameter comparison between the two 3D structure models have been shown in Table 5. And we have given explanation in Line 182-186.

The accession number of protein sequence has been added. (Line110)

Reviewer 2 Report

Specific comments

A 2 decimal points are sufficient in general, for THI one figure after decimal point sufficient

Meteorological data are they maximum, minimum, mean (max+mim/2) specify

Line

127-9                    “all dominant  genotypes related to heat-tolerance in Zebu population”  support content

157=163              “there was no significant difference between individuals”  here and In other places, new unknown statistics ?

170-3                    heat tolerance  traits: individuals with Hap 1/1 was the most potentially related to  heat tolerance traits. Sentence not clear

197-202               logical inference not clear

225-6                    sentence not clear

General comments

No data were brought to support the claim of higher heat tolerance in particular groups

Annual mean climate data do not indicate heat stress presence

Between regions differences in frequencies of genetic constitution might be due to animal type tradition due preferences.

Author Response

Dear reviewer

Thank you very much for your suggestions on my manuscript. For your question, my reply is as follows:

Point 1: A 2 decimal points are sufficient in general, for THI one figure after decimal point sufficient

Meteorological data are they maximum, minimum, mean (max+mim/2) specify

Response 1:

All the Numbers have been modified

Point 2: Line127-9                    “all dominant  genotypes related to heat-tolerance in Zebu population”  support content

157=163              “there was no significant difference between individuals”  here and In other places, new unknown statistics ?

Response 2:

Line 127-9  We described that the individuals with mutant genotypes for the four loci are widely distributed in the Zebu population which has strong heat adaptability, while the individuals with wild genotypes have a wider distribution in the Angus population which has weak heat adaptability. These are obtained by analyzing the  frequencies in Table S2.

Line 157-163  In this paragraph, we showed the results of the association analysis between SNPs and environmental parameters. And the detailed data was shown in Table 3.

Point 3: 170-3                    heat tolerance  traits: individuals with Hap 1/1 was the most potentially related to  heat tolerance traits. Sentence not clear

Response 3:  

Modified as: “individuals with Hap 1/1 was the most potentially related to heat tolerance traits.”

Point 4: 197-202               logical inference not clear

Response 4:  

Modified as: “With the advent of escalating environmental stress due to climate change, the challenging tropical stress could have determine more favourable genotypes for bovine [22]. Nevertheless, as the environments and management intervention strategies implementation, environmental regulation removed environmental stressors, this would make natural selection be adversely affected. In consequence, screening of candidate gene related to heat tolerance and breeding of heat-resistant cattle are of great significance to the production of cattle.”

Point 5: 225-6                    sentence not clear

Response 5:  

Modified as: “This trend was consistent with the distribution of different climatic conditions in China (Table S1). It also shows an obvious distribution trend from south to north. “

Point 6: No data were brought to support the claim of higher heat tolerance in particular groups.

Annual mean climate data do not indicate heat stress presence.

Between regions differences in frequencies of genetic constitution might be due to animal type tradition due preferences.

Response 6:  

We have selected two breeds with known heat-adaptation as a control group, Zebu (distributed in tropical) and Angus (distributed in cold environment). Ref: Hansen P J. Physiological and cellular adaptations of zebu cattle to thermal stress. Animal Reproduction Science. 2004, 82(4): 349-360.

“McDowell et al. [18] suggested that temperature-humidity index (THI) can be used as an indicator to ascertain heat load intensity of thermal climatic conditions by measuring the combined effects of annual mean temperature (T) and relative humidity (H). Solar radiation was affected by solar altitude angle, local climate types and solar radiation time, which had the largest influence (r2=0.92), and solar radiation time basically reflected the solar radiation amount [19].” (Line108-112). These could prove four climate indicators we chose are effective. Firstly, Annual mean climate data can largely reflect the climate condition of an area. What’s more, beef cattle are generally raised in semi-open or even fully open environments in China. So cattle are affected by the local climate. And the climate data we used were the mean of the data over the last 30 years, these could sufficiently represent the long-term natural selection conditions. Moreover, our study used 1072 individuals from 36 breeds, it is a very large sample size to show the association.

The samples in this study were all from ordinary beef cattle rather than special varieties for other special purposes. And from the perspective of quantitative genetics, the effect of environmental factors on breeding can reach more than 50%. Animal preferences can be understood as the result of long-term adaptation to the environment.

In addition, other studies using the same methodology have been published online.

Ref: Zeng L, Cao Y, Wu Z, et al. A Missense Mutation of the HSPB7 Gene Associated with Heat Tolerance in Chinese Indicine Cattle. Animals (Basel). 2019;9(8):554. Published 2019 Aug 14. doi:10.3390/ani9080554

Wang K, Cao Y, Rong Y, et al. A Novel SNP in EIF2AK4 Gene Is Associated with Thermal Tolerance Traits in Chinese Cattle. Animals (Basel). 2019;9(6):375. Published 2019 Jun 19. doi:10.3390/ani9060375

Zeng, L. , Chen, N. , Ning, Q. , Yao, Y. , Chen, H. , Dang, R. , Zhang, H. and Lei, C. (2018), PRLH and SOD1 gene variations associated with heat tolerance in Chinese cattle. Anim Genet, 49: 447-451. doi:10.1111/age.12702

Reviewer 3 Report

I suggest to revise the english carefully.

Author Response

Dear reviewer

Thank you very much for your suggestions on my manuscript. We regret there were problems with the grammar. The paper has been carefully revised to improve the grammar and readability.